# Optimization of the Biotreatment of GTL Process Water Using *Pseudomonas aeruginosa* Immobilized in PVA Hydrogel

Somaya A. Ahmed [ID] , Riham Surkatti, Muneer M. Ba-Abbad [ID] and Muftah H. El-Naas *[ID]

Gas Processing Center, College of Engineering, Qatar University, Doha P.O. Box 2703, Qatar
* Correspondence: muftah@qu.edu.qa

**Abstract:** The COD reduction in gas to liquid (GTL) process water was optimized using response surface methodology (RSM). The biodegradation process was carried out in a spouted bed bioreactor (SBBR) using *Pseudomonas aeruginosa* immobilized in polyvinyl alcohol (PVA) gel. Different factors affecting the biological treatment of GTL process water (PW) were investigated. Three variables including PVA volume fraction, initial COD, and pH were investigated in the batch experiments. The biodegradation experiments were carried out by varying the initial COD values from 1000 to 3000 mg/L, pH from 5 to 8, and PVA v% from 20 to 30%. The maximum COD reduction was estimated to occur at an initial COD of 2595 mg/L, PVA v% of 27%, and pH of 7.3. At optimum conditions, the bioreactor system was able to achieve a maximum COD reduction of 89%, which is quite close to the RSM prediction value of 90%. The optimum operating conditions were used to carry out continuous biodegradation, and the results indicated that the COD reduction increased from 60% to 62% with an increase in the air flow rate from 2 to 3.3 $L_a/L_r$.min. However, by increasing the liquid flow rate from 2.1 to 4.2 mL/min and back to 2.1 mL/min, the COD reduction decreased from 66% to 39%. The system responded quickly to the change in liquid flow rate and returned to the initial COD level. This indicates that the system is highly stable and can easily recover.

**Keywords:** GTL process water; biodegradation; spouted bed bioreactor (SBBR); *Pseudomonas aeruginosa*; response surface methodology (RSM)





## 1. Introduction

The need for environmentally friendly fuels for energy has recently been steadily increasing. Gas-to-liquid (GTL) is a technological breakthrough that uses the Fischer–Tropsch (FT) process to transform natural gas (NG) into high-performance, ultra-clean liquid fuels. This innovative and rising technology is likely to contribute to a higher proportion of global gas processing in the future. The development of GTL technology has advanced technologically over the past ten years, and many commercial-scale plants have been constructed across the world [1]. GTL process water (PW) is a by-product of the Fischer–Tropsch (FT) reaction that produces water in significant quantities (~25%), more than other hydrocarbon products on a weight basis. Process water produced by GTL industries must be treated to meet the regulatory agency standards for both safe releases into water bodies and effective reuse [2]. The majority of COD in the GTL PW water stream comes from alcohols, aldehydes, and ketones, which can be effectively treated biologically.

Biological approaches are popular in the wastewater treatment field because of their ease of use, environmental friendliness, low cost, and long-term application [3]. Biological approaches involve using organisms such as bacteria, fungi, and algae to mineralize or break down contaminants into less toxic compounds [4]. The use of bacteria is beneficial since these microorganisms are flexible representatives and contain a variety of Actinomycetes [5]. Furthermore, these bacteria can produce spores, resist different contaminants, and survive in several environments. The organic pollutants in the GTL PW can be effectively treated biologically under aerobic and anaerobic conditions. The course of treatment

may also combine anaerobic and aerobic systems. Several studies on GTL PW, namely FT wastewater, have been investigated on the bench and laboratory scales using both synthetic and actual wastewater, and biological treatment has been deliberated under anaerobic conditions [6]. GTL PW is usually treated using a well-known biological process in conventional activated sludge (CAS) systems. This method of treatment is based on biomass, which is held by a settler, aerobically degrading organic contaminants. Three steps of GTL PW treatment, including chemical, biological, and physical treatment approaches, were described by Pon Saravanan and Van Vuuren [2]. The three integrated processes of the GTL PW treatment plant used chemical treatment in the first stage to remove free oil and suspended hydrocarbons, followed by biotreatment in an aeration tank to eliminate carbonic and nitrogenous compounds, and finally physical treatment such as sand filtration in the third stage to eliminate suspended solids, oil, chemical oxygen demand, and related biological oxygen demand [2]. The CAS process was combined with ultrafiltration (CAS-UF) and used in the treatment of GTL PW. The CAS effluent can undergo post-treatment ultrafiltration (UF) to remove unsettled particles and further lower the COD in the effluent, allowing for the reuse of the treated water [7]. Additionally, Fischer–Tropsch (FT) reaction water from gas-to-liquid (GTL) industries was also treated using a membrane bioreactor (MBR) system and compared to the current treatment system [7]. Majone et al. [8] investigated the anaerobic biodegradation of synthetic FT wastewater with a high concentration of COD (~28 g/L) generated by long-chain alcohols utilizing a continuous flow-packed bed biofilm reactor (FPBBR) on a laboratory scale. They steadily increased the COD content in tests to evaluate the inhibitory effect of long-chain alcohol concentrations and achieved about 96% of COD reduction. Aerobic degradation of FT wastewater was studied by Chain et al. [9] to decrease the high COD specifically from short-chain alcohols and volatile fatty acids as they represented around 87% of the given wastewater. The FT wastewater was synthesized from SCAs and VFAs in a mineral salt solution and a COD of 67.9 g/L and biodegraded using *Bacillus* sp. Within 3 days, the strains reduced COD by up to 90% and effectively degraded the organic pollutants. Other studies [6,10] have suggested a combination of anaerobic biological treatment and chemical strategies to address the issue of incomplete degradation for long-chain alcohol under high organic load. Bio-electrochemical systems (BES) have been used in the treatment of FT wastewater by utilizing electrochemically active bacteria as a catalyst for oxidation and/or reduction processes at the anode and/or the cathode [6]. By combining an anaerobic digester with a BES, the treatment performance was improved as well as the production of biogas [11].

In most biological treatment processes of GTL PW, activated sludge is used as a biomass source where the biological community, considered as a mixed culture, contains a wide range of bacteria, fungi, and algae. Despite the biodegradability of these microorganisms, several types of pure cultures have been used over the years for the biodegradation of organic contaminants. *Pseudomonas* was first identified as a Gram-negative, polar-flagellated, and rod-shaped bacteria by Migula in 1894, during the 19th century [12]. It is one of the prokaryote genera that has undergone the most research (bacteria). *Pseudomonas aeruginosa* is a Gram-negative bacterium that can be found in almost any environment. Their metabolic capacity is vast, as evidenced by their ability to synthesize a wide range of secondary metabolites and polymers as well as their ability to employ a wide range of carbon sources and electron acceptors [13]. Many researchers have studied the ability of *P. aeruginosa* to degrade various chemicals in the batch and continuous bioreactors under various conditions. These include the degradation of 0.02% naphthalene [14], the biodegradation of volatile organic compounds [15], the biodegradation of petroleum compounds [16], the degradation of octamethylcyclotetrasiloxane (D4) [17], and the bioremediation of heavy metals [18]. However, the application of *P. aeruginosa* for the biotreatment of GTL PW has never been previously documented. Hence, this study aimed to investigate the biodegradation of GTL water by *P. aeruginosa* bacteria using a laboratory-scale spouted bed bioreactor system (SBBS). Moreover, the interaction between the operating parameters and the optimal conditions for the maximum reduction efficien-

cies by varying test parameters such as the initial COD concentration, PVA v%, and pH using RSM was investigated.

## 2. Materials and Methods

### 2.1. Chemicals and Materials

Most of the chemicals including NaOH pellets for pH adjustment and mineral salts including $MgSO_4 \cdot 7H_2O$, $K_2HPO_4$, $CaCl_2 \cdot 2H_2O$, $(NH_4)_2CO_3$, $FeSO_4 \cdot 7H_2O$, $ZnSO_4 \cdot 7H_2O$, $MnCl_2 \cdot 4H_2O$, $CuSO_4 \cdot 5H_2O$, $CoCl_2 \cdot 6H_2O$, and $Na_2MoO_4 \cdot 2H_2O$ were obtained from Sigma Aldrich, St. Louis, MO, USA. However, PVA powder was obtained from BDH, London, UK.

### 2.2. GLT PW Samples

The water samples were obtained from a local GTL plant in Qatar and pretreated using air stripping to remove volatile organic pollutants. GTL PW is characterized by high acidity and COD content. Table 1 shows the physical and chemical properties of the original and pretreated GTL process.

**Table 1.** The physical and chemical characteristics of GTL.

| Characteristic | GTL PW | Pretreated GTL PW |
|---|---|---|
| COD (mg/L) | 5000–7000 | 2000 to 4000 |
| TOC (mg/L) | 1500–1700 | 700–1400 |
| pH | 2.9 | 3.3 |

### 2.3. Isolation and Immobilization of Bacterial Culture

*Pseudomonas aeruginosa* was isolated with other hydrocarbon-degrading bacteria from different highly contaminated soils in Qatar. According to Al Disi et al. [19], at a tillage depth of 1–2 cm, random sampling was taken with a sterile spatula from various spots. The soil samples were collected and placed in clean glass bottles, which were then securely sealed, labeled, and twisted with foil to prevent contamination and prevent any further light reactions. The technique for enrichment, isolation, identification, and studying the activation of hydrocarbon-degrading bacteria was described thoroughly by Al Disi et al. [19]. The isolated *Pseudomonas aeruginosa* bacterial cells were collected in a glass jar and kept refrigerated until they were immobilized in PVA gel.

To prepare PVA with 10 wt%, a homogenous PVA solution was made by mixing 100 g of PVA powder with 850 mL of distilled water and 50 mL of a suspension of the isolated bacteria cells at roughly 70–80 °C. The solution was then mixed using a glass rod to ensure its homogeneity. PVA solution was poured into ice molds and cross-linked using freezing–thawing cycles, by freezing at −20 °C for 20 h, and thawing at +20 °C for 4 h. The FT process was repeated four times. This prepared PVA gel is known to produce a high-porosity polymeric matrix with good mechanical strength and stability [20].

### 2.4. Biomass Acclimatization

The immobilized bacteria were slowly acclimatized to GTL PW by placing them in a batch SBBR with GTL PW and a mineral nutrient solution. The acclimatization process was performed by a gradual increase in the GTL PW COD in the range from 500 to 2000 mg/L over a period of two weeks. After this step, the bacteria were fully acclimatized to GTL-PW and were ready for the biodegradation process [20]. Table 2 shows the composition of mineral salts added to the GTL PW solution.

**Table 2.** Composition of the mineral salt medium [21].

| Component | Concentration (mg/L) |
| --- | --- |
| $MgSO_4 \cdot 7H_2O$ | 300 |
| $K_2HPO_4$ | 250 |
| $CaCl_2 \cdot 2H_2O$ | 150 |
| $(NH_4)_2CO_3$ | 120 |
| $FeSO_4 \cdot 7H_2O$ | 3.5 |
| $ZnSO_4 \cdot 7H_2O$ | 1.3 |
| $MnCl_2 \cdot 4H_2O$ | 0.13 |
| $CuSO_4 \cdot 5H_2O$ | 0.018 |
| $CoCl_2 \cdot 6H_2O$ | 0.015 |
| $Na_2MoO_4 \cdot 2H_2O$ | 0.013 |
| Total | 824.98 |

*2.5. Spouted Bed Bioreactor System (SBBR)*

The SBBR utilized in this experiment was made of Plexiglas and had a total volume of 1.5 L. The SBBR is characterized by systematic intensive mixing caused by the cyclic motion of particles within the bed, which is caused by a single air jet injected through an aperture at the bottom of the reactor [22]. The temperature (at 32 °C) of the bioreactor was controlled through the circular movement of the water in the water jacket surrounding the reactor. Air was continuously supplied into the reactor at a certain flow rate to improve mixing and to also provide oxygen to maintain aerobic conditions [23,24]. A schematic diagram of the SBBR is shown in Figure 1.

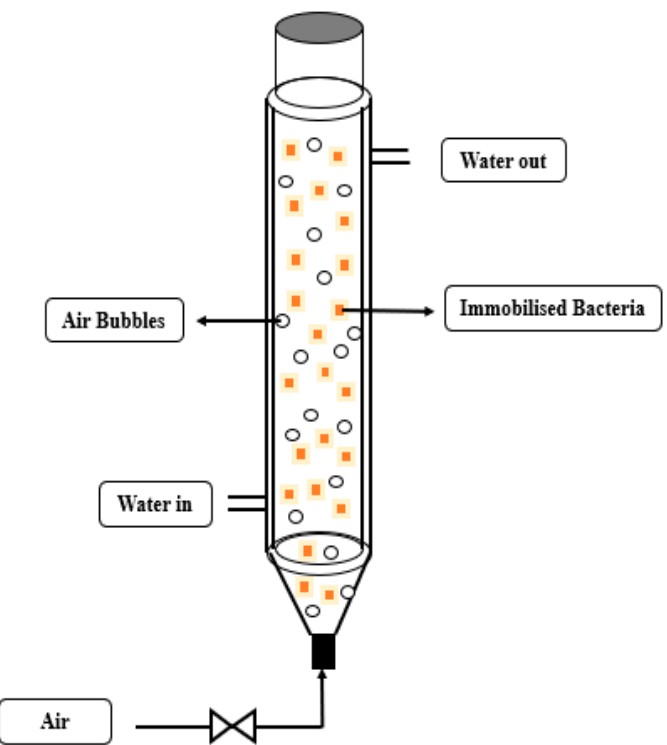

**Figure 1.** A schematic diagram of the spouted bed bioreactor (SBBR).

*2.6. Batch Biological Treatment of GTL PW*

In batch experiments of GTL PW, the reactor was initially filled with the standard nutrient medium solution as well as the biocatalyst, which is the PVA gel with the immobilized bacteria, which occupied 20 to 30% of the reactor volume based on previous studies [22]. The reactor was filled with a total volume of 1.5 L (GTL PW and immobilized bacteria). Air was injected into the conical bottom of the bioreactor to ensure proper mixing

and interaction between the substrate and the bacterial cells within the bioreactor as well as to provide the oxygen required for biodegradation. The air flow rates, and liquid flow rates were adjusted to 3.3 $L_a/L_r$.min and 2.4 mL/min, respectively. The reactor temperature was held at almost 32 °C. This temperature was found to be optimum in a past study [23], and is in good agreement with the values mentioned in the literature [24]. The initial pH solution was kept at pH 7 using NaOH pellets as these conditions are regarded to be the best for organic biodegradation according to El-Naas et al. [22].

### 2.7. Continuous Biological Treatment of GTL PW

Continuous experiments were performed to study the effects of air flow rates at 2, and 3.3 $L_a/L_r$.min, and liquid flow rates at 2.1 and 4.2 mL/min on the biotreatment of GTL PW under optimum conditions selected based on RSM. To study the response of the SBBR system to other operating factors including air and liquid flow rates, continuous experiments were carried out at constant COD of 2200 mg/L, pH of 7.29, PVA v% of 27%, and temperature of 32 °C. The stripped GTL PW mineral solution was constantly introduced to the reactor during all experimental runs, utilizing a peristaltic pump at a stable liquid flow rate with an accuracy of ±1 mL/min. The samples were collected and analyzed using the COD at various periods. All continuous experiments were carried out in replicates and the average was plotted. The standard error ranged from 5 to 10% of the reported average.

### 2.8. Analytical Methods

A HAC-UV spectrophotometer with COD reagents was used to conduct the COD analysis. Two mL of the water sample was added to the HAC LCK514 cuvettes and heated for 2 h to complete the reaction between the reagent and the water sample. The COD content in mg/L was measured using the HAC DR 3900.

### 2.9. Statistical Analysis and Optimization Using RSM

The current study aimed to investigate the biotreatment of GTL PW by *Pseudomonas aeruginosa* bacteria as well as the interaction between operating parameters and the optimum conditions for the maximum COD reduction efficiencies by varying the test parameters such as the initial COD concentration, immobilized bacteria volume percentage, and pH using RSM. The RSM was carried out for the biotreatment of the two types of wastewater under specific experimental conditions. The parameters were selected based on the screening experiments. Using RSM, a large amount of data can be obtained with a small number of experimental operations, and there are many important advantages such as the ability to determine not only the effects of individual parameters, but also their relative importance in a given and the interactive effects of two or more variables. Table 3 lists the three variables used in this study, each of which has two levels (upper and lower).

**Table 3.** Variables and levels applied for the removal of COD in GTL PW.

| Factor | Units | −α | Lower Limit (−1) | 0 | Upper Limit (+1) | +α |
|---|---|---|---|---|---|---|
| Concentration | mg/L | 318.21 | 1000 | 2000 | 3000 | 3681.79 |
| pH | - | 3.98 | 5 | 6.5 | 8 | 9.02 |
| PVA v% | - | 16.59 | 20 | 25 | 30 | 33.41 |

The central composite method (CCD) experimental design of RSM was chosen to find the relationship between the response functions and variables using the statistical software tool MINITAB 20. In the central composite method, a total number of 20 experiments including center points was augmented with a group of axial points (also called star points) to estimate the curvature.

The reduction rate of COD was determined according to the following equation:

$$\text{COD Reduction } \% = \frac{C_0 - C}{C_0} \tag{1}$$

where C (mg/L) is the final COD concentration of the sample, and $C_0$ (mg/L) represents the initial COD of the water sample.

Using the optimal design factor (RSM), the coefficients were also predicted in a quadratic polynomial mathematical model to predict the COD reduction efficiencies. Table 4 lists the several parameter combinations used in the central composite design by response surface methodology using CCD.

**Table 4.** The several parameter combinations used in the design of the GTL removal experiments.

| RunOrder | PtType | Blocks | COD | pH | PVA Vol. % |
|---|---|---|---|---|---|
| 1 | 0 | 1 | 2000 | 6.5 | 25 |
| 2 | 1 | 1 | 3000 | 8 | 20 |
| 3 | 1 | 1 | 1000 | 5 | 30 |
| 4 | −1 | 1 | 2000 | 3.98 | 25 |
| 5 | −1 | 1 | 2000 | 6.5 | 16.59 |
| 6 | 1 | 1 | 3000 | 5 | 20 |
| 7 | 1 | 1 | 3000 | 5 | 30 |
| 8 | 0 | 1 | 2000 | 6.5 | 25 |
| 9 | −1 | 1 | 2000 | 9.02 | 25 |
| 10 | −1 | 1 | 3681.79 | 6.5 | 25 |
| 11 | −1 | 1 | 318.20 | 6.5 | 25 |
| 12 | 1 | 1 | 1000 | 8 | 20 |
| 13 | 0 | 1 | 2000 | 6.5 | 25 |
| 14 | 0 | 1 | 2000 | 6.5 | 25 |
| 15 | 0 | 1 | 2000 | 6.5 | 25 |
| 16 | 1 | 1 | 1000 | 8 | 30 |
| 17 | −1 | 1 | 2000 | 6.5 | 33.40 |
| 18 | 1 | 1 | 1000 | 5 | 20 |
| 19 | 0 | 1 | 2000 | 6.5 | 25 |
| 20 | 1 | 1 | 3000 | 8 | 30 |

## 3. Results and Discussion

### 3.1. Statistical Analysis

All statistical combinations of the variables were evaluated using the DOE software, which yielded the design outcomes from the experiments. *P. aeruginosa* had high coefficients of determination R2 of 0.8140 for the biotreatment of GTL PW. RSM was used to optimize the degradation conditions. The CCD method was used to investigate the effects of the initial COD concentration, PVA v%, and pH on COD reduction. Equation (2) shows the quadratic polynomial mathematical model to predict the COD reduction efficiencies.

Regression Equation in Uncoded Units:

$$\begin{aligned}\text{COD Reduction } \% = {} &-304.5 + 0.0198\,\text{COD} + 37.0\,\text{pH} + 17.29\,\text{PVA v}\% - 0.000004\,\text{COD} * \text{COD} - 1.654\,\text{pH} * \text{pH} \\ &-0.2728\,\text{PVA v}\% * \text{PVA v}\% - 0.00077\,\text{COD} * \text{pH} + 0.000151\,\text{COD} * \text{PVA v}\% - 0.403\,\text{pH} \\ &* \text{PVA v}\%\end{aligned} \tag{2}$$

### 3.2. Effect of Initial COD

The contaminant concentration is one of the most important factors that affect the biological treatment of wastewater. The increase in the organic concentration may inhibit the biodegradation process or increase the degradation efficiency [25]. Figure 2a,b shows the effect of interaction between the initial COD and other two process parameters, namely, pH, and PVA volume fraction on the removal of COD. At a constant pH of 5, a PVA v% of 20, and an initial COD of 0f 1000 mg/L, the COD reduction % was 58. However, under the

same conditions, increasing the initial COD to 3000 mg/L resulted in a COD reduction % of 69.89 (around a 12% increase). Similar behavior was also noticed at a constant pH of 5, and PVA v% of 30, where an increase of the initial COD from 1000 mg/L to 3000 mg/L resulted in a COD reduction from 62.8% to 83.27%, respectively (approximately 21% increase). This behavior reveals that when the COD levels are higher, there is a greater oxygen demand. This implies that water with high COD levels likely contains more oxidizable organic material, and the microorganism can adapt to the higher organic concentration, resulting in high removal efficiency [26]. Moreover, one can observe that the biodegradation increased as COD increased from 500 to ~2600 mg/L and then decreased. It is anticipated that exposure to a high concentration of contaminants might suddenly have detrimental effects on the bacterial enzymes that are often responsible for the major stages in the biodegradation process [27]. This also implies that wastewater with high COD levels has lower dissolved oxygen (DO) values. Life forms need oxygen to survive, hence a low quantity of dissolved oxygen is dangerous. Therefore, it is advisable to increase DO concentrations by reducing COD levels in wastewater before releasing it [26].

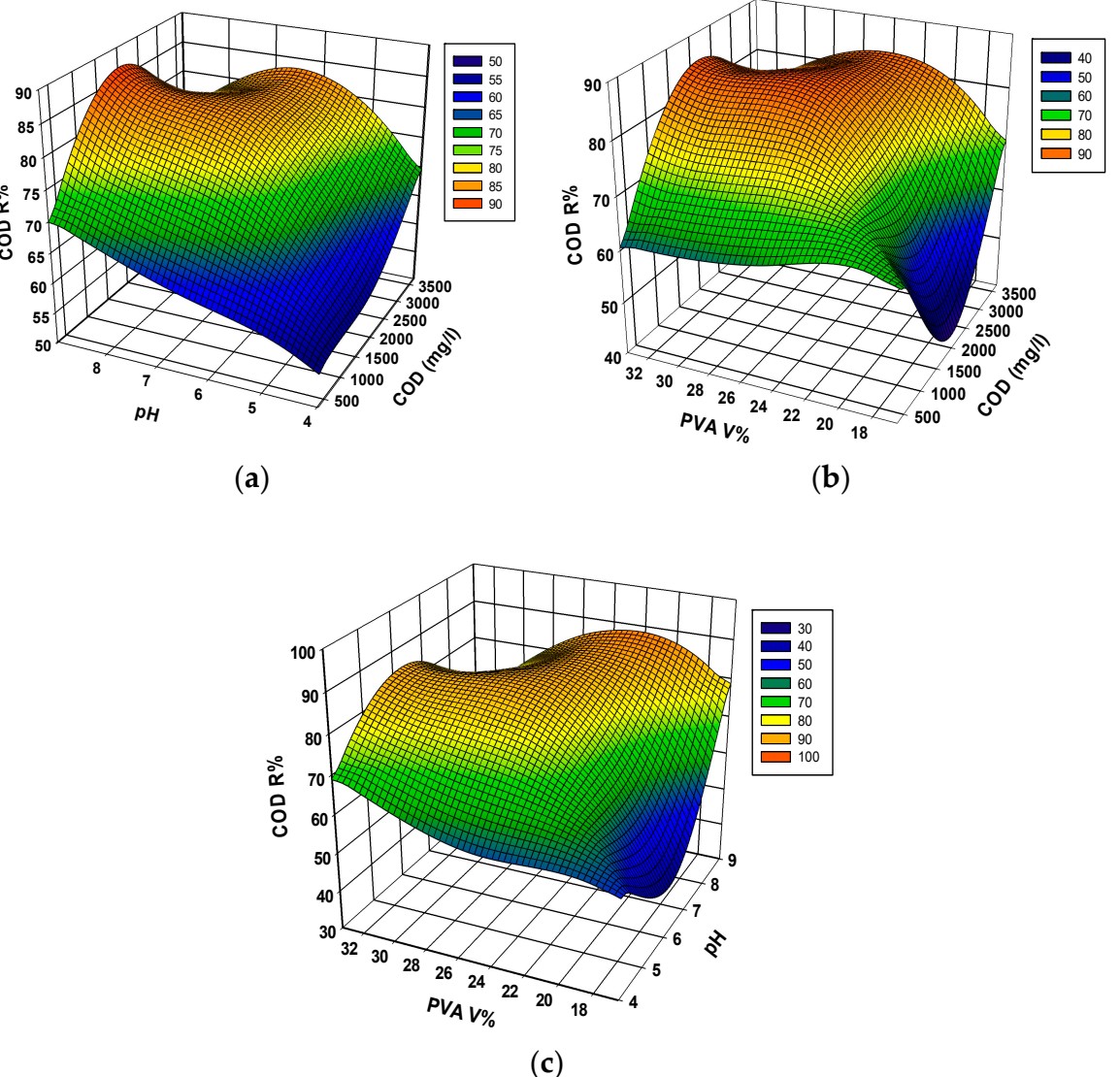

**Figure 2.** The interaction between the experimental parameters in the GTL experiment. (**a**) COD reduction % vs. pH and initial COD (mg/L); (**b**) COD reduction % vs. initial COD concentration (mg/L) and PVA volume fraction; (**c**) COD reduction % vs. pH and PVA volume fraction.

### 3.3. Effect of pH

The initial pH of the solution has a big impact on the microbial growth and enzyme activity, therefore, it is significant when it comes to developing biological treatment methods. Similarly, the effects of the interaction relation between pH with the initial COD and PVA volume fraction on the COD reduction rate were determined and the results are shown in Figure 2a,c. At a constant initial COD of 2000 mg/L, a PVA v% of 20, and an initial pH of 3.9, the COD reduction % was 58.15. However, under the same conditions, increasing the pH to 6.5 resulted in a COD reduction % of 86.77 (around a 29% increase). Because almost all biological species seemed to have optimal pH conditions, pH has a physiological effect on microbial activity, similar to temperature [28]. Although some species need a restricted pH range, others are more tolerant of a larger pH range. Some species thrive in high-pH environments, whereas others thrive in low-pH environments. Research of *Pseudomonas aeruginosa* biotreatment of GTL-produced water mixtures found that biomass activity was entirely reduced at pH 5, 9, and 10, with pH 6–8 being the optimal [28]. Because most bacteria are neutrophils, the optimal pH for the most efficient biotreatment of organic substances is usually about 7.5 [29].

### 3.4. Effect of PVA Volume %

The rate of biodegradation of organic pollutants is significantly influenced by the volume fraction of PVA pellets, which is directly related to the quantity of active biomass cells in the bioreactor [30]. Figure 2b,c show the effect of interactions between PVA v% with initial COD concentration, and pH respectively. The number of active biomass cells in the bioreactor is directly proportional to the volume fraction of PVA pellets, and so plays an essential role in determining the rate of COD reduction. At a constant initial COD of 2000 mg/L, pH of 6.5, and PVA v% of 16.9, the COD reduction % was 42.85. However, under the same conditions, increasing the PVA v% to 30 resulted in a COD reduction % of 86.77 (around a 44% increase). The COD reduction increased as the PVA volume fraction increased. However, the removal effectiveness of PVA v% 25 and 33% appeared to be nearly the same. This could be because 33% had a greater biomass and 25% had better mixing. It is believed that increasing the volume fraction of PVA above 30% is likely to result in reduced mixing and, as a result, lower degradation performance [30].

The optimization of the biodegradation of the GTL PW was evaluated. Figure 3 shows that the optimum conditions obtained from RSM to achieve a maximum COD reduction efficiency of 90% was predicted to occur at an initial COD of 2595 mg/L, PVA volume fraction of 27%, and pH of 7.3.

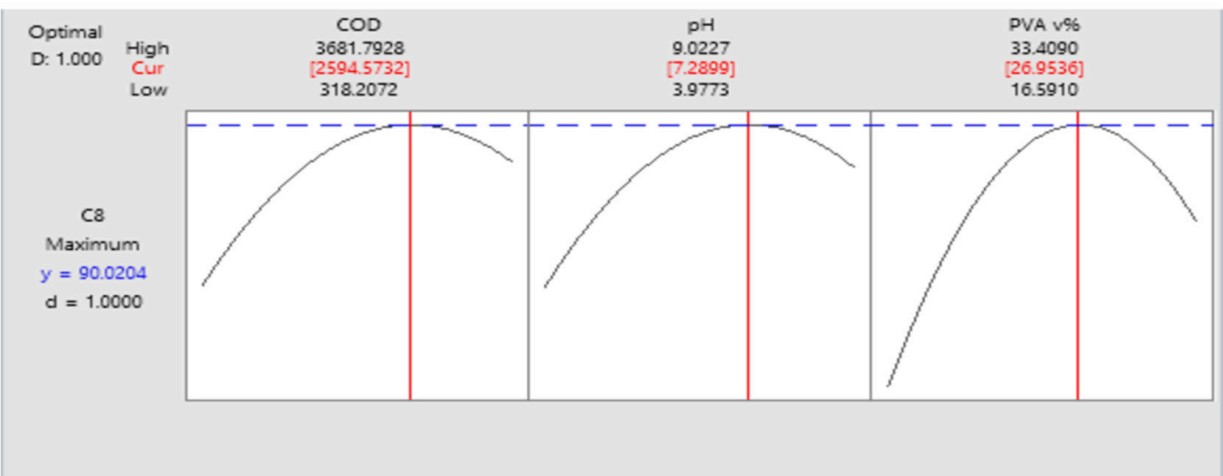

**Figure 3.** The optimum conditions for COD reduction % were obtained using the response optimizer.

### 3.5. Model Validation

The experimental response (i.e., COD reduction %) was compared to the predicted response to validate the accuracy of RSM, as shown in Figure 4. The collected experimental values, illustrated by scattered points, showed a linear relationship between the experimental and predicted values. The $R^2$ value of 0.8481 indicates a close fit between the modeled and collected data. Moreover, the biodegradation experiment was carried out at optimum conditions obtained from the optimization of the COD reduction to the maximum value. Results showed that the predicted optimal COD from RSM had a good match to the experimental value (90% and 89%, respectively). This means that the model for the biotreatment of GTL PW was valid.

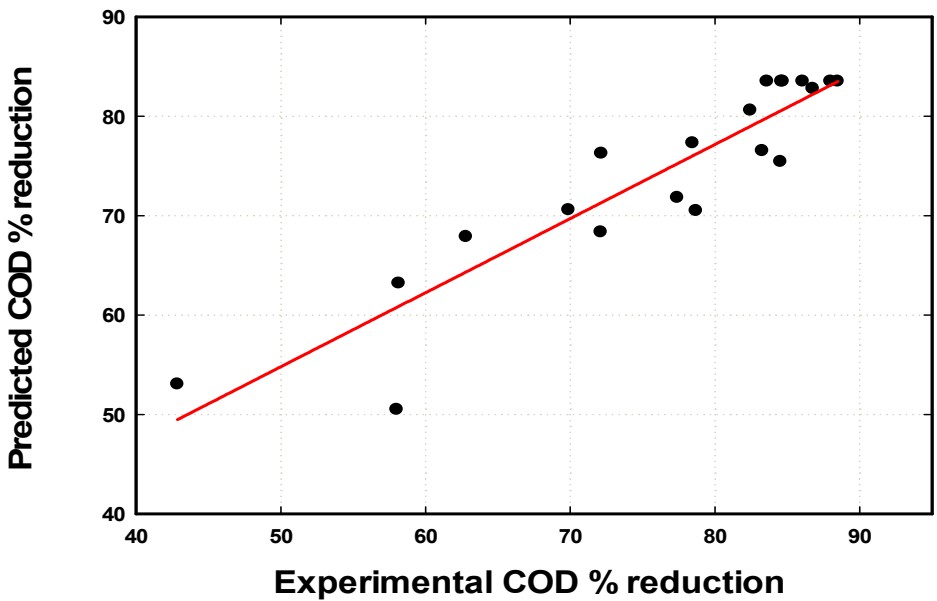

**Figure 4.** Predicted versus experimental COD reduction values for GTL PW.

### 3.6. Continues Biological Treatment of GTL PW

3.6.1. Effect of Air Flow Rate

The bioreactor's air flow rate is crucial in ensuring that there is enough oxygen for biodegradation and adequate mixing via particle movement. Different air flow rates of 1, 2, and 3.3 $L_a/L_r$.min (liter of air per liter of the reactor per min) were operated for 24 h to evaluate the impact of air flow rate on the continuous biological treatment of GTL PW. However, at 1 $L_a/L_r$.min, the aeration and mixing of the immobilized bacteria were limited, thus, only 2, and 3.3 $L_a/L_r$.min were studied. The initial COD concentration, liquid flow rate, and temperature were kept at the optimum conditions of ~2200 mg/L, 2.1 mL/min, and 32 °C, respectively. Samples were collected from the effluent for the COD analysis. Figure 5 shows COD reduction with increasing time for two different air flow rates 2, 3.3 $L_a/L_r$.min. When comparing the airflow rates of 2 and 3.3 $L_a/L_r$.min, the difference in DO was found to be insignificant (5.2, and 5.5 mg/L, respectively). Moreover, there was a slight difference in the mixing of immobilized bacteria between the two flow rates, which could be attributed to the lower COD reduction at 2 $L_a/L_r$.min (60.6%) compared to the air flow rate of 3.3 $L_a/L_r$.min (62%). The findings demonstrate that the air flow rate affects the biodegradation rate of organic pollutants. In general, at a specific range, the higher the airflow rate, the better the biodegradation rate, which are two key factors for how the air flow rate affects the biotreatment of organic compounds by feeding the necessary amount of oxygen [21]. Moreover, Gopalakrishnan et al. [31] reported that the excess supply of oxygen led to the higher degradation activity of the biomass, therefore, the increase in airflow rate will increase the reduction in COD. However, at a lower range, the biodegradation rate will decrease due to the fact that, after a certain limit, critical velocity

occurs, causing the biomass support particles to settle down at the bottom of the reactor. As a result, there are fewer interactions between the substrate, biomass, and air, which leads to less reduction in COD. For the air flow rate at a lower range, oxygen is the growth-limiting factor [31].

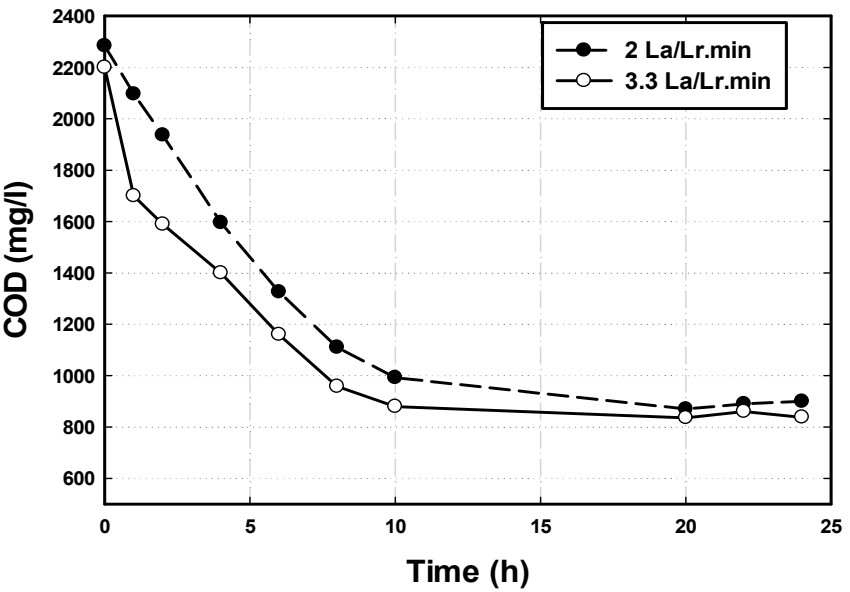

**Figure 5.** The concentration of GTL PW in SBBR vs. time for two different air flow rates (2, and 3.3 $L_a/L_r$). Initial COD = ~2200 mg/L; PVA volume % = 27; reactor temperature = 32 °C; liquid flow rate = 2.1 mL/min.

### 3.6.2. Effect of Liquid Flow Rate

To assess the effect of liquid flow rate step change on the biotreatment of GTL PW, a continuous experiment was carried out at the optimum conditions obtained from batch experiments in Section 3.4 (initial COD of ~2200 mg/L; PVA v% of 27; reactor temperature at 32 °C; and an airflow rate of 2.1 L/min and HRT of 12 h). Figure 6 shows the decrease in COD reduction through these step changes in the liquid flow rate. First, the experiment was carried out for around 22 h to reach a steady state and a COD reduction of 65.69% was achieved. Then, a step change of liquid flow rate from 2.1 to 4.2 mL/min was performed, reducing the HRT to 6 h. A second steady state condition with a liquid flow rate of 4.2 mL/min was reached, where the COD reduction declined to 38.61%. After this time, the liquid flow rate was returned to its previous value to reach the third steady state condition with a 63.68% COD reduction. This is to be expected because a higher liquid flow rate reduces the residence time in the bioreactor HRT from 12 h to 6 h at liquid flow rates of 2.1 and 4.2 mL/min, respectively. Thus, the contact time between immobilized bacteria and organic pollutants PW reduced and resulted in lower COD reduction [21]. It is generally accepted that the inlet feed flow rate has the greatest influence on determining the efficiency of bioreactors by governing the retention time of the contaminants within the bioreactor [20,32]. The more the HRT, the greater the chance for the efficient decomposition of the organic contaminant. A very low liquid flow rate, however, may also result in mass transfer constraints, which could lead to a decrease in pollutant removal [33]. By increasing the liquid flow rate, more organic pollutants are introduced to the bioreactor, thus the substrate may serve as a nutrient for the microbial biomass until a certain initial concentration, however, an increase in substrate concentration may impart the toxic effect of pollutants on the metabolic activity of microbial biomass, resulting in lower removal efficiency [33].

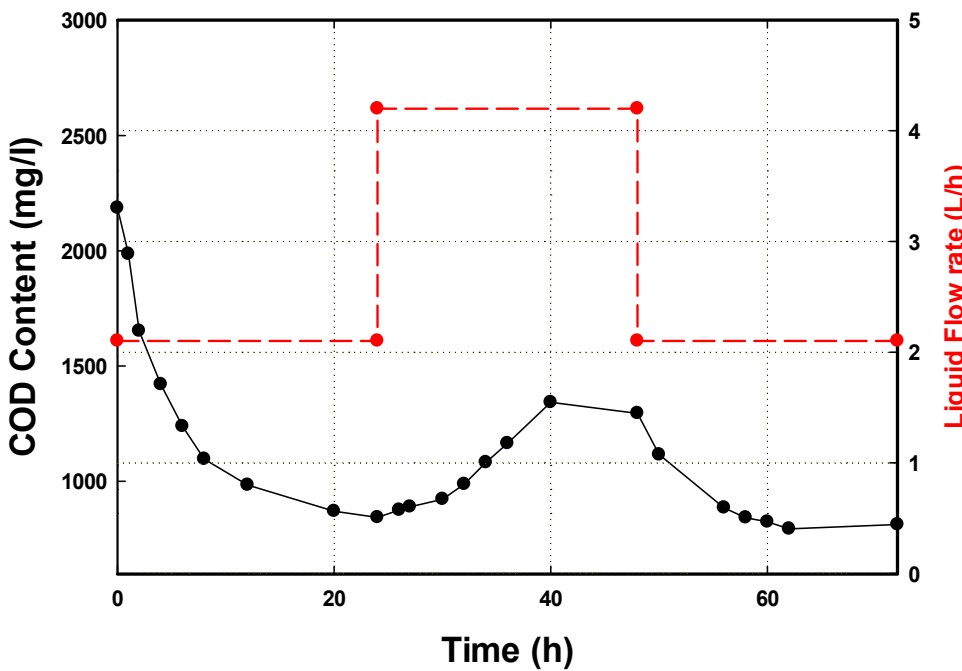

**Figure 6.** Liquid flow rate step-change of (2.1–4.5 and back to 2.1 mL/min). Initial COD = 2200 mg/L; temperature = 32 °C; air flow rate 3.3 $L_a/L_r$.min.

Kawan et al. [34] reported that the water flow or movement in the reactor caused a high water velocity in the reactor, a short duration of contact between the organic matter, and less time for biofilm formation when the reactor was run at a high flow rate (short retention time (HRT)). As a result, the capability of the organisms to metabolize organic materials was diminished. Additionally, Chen et al. noted that minimizing the hydraulic retention time (HRT) and lowering the pH level in the system are two ways to minimize the excessive accumulation of butyric acid and propionic acid, which is a critical component of successful FT wastewater treatment [35].

### 3.6.3. Dynamic Behavior

The variation in the load, the hydraulic characteristics of the reactor (transport and mixing), and the transformation processes are all factors that affect the dynamic, time-dependent behavior of reactors. Many effects are unreasonable and cannot be fully understood without thoughtful deliberation and analysis. A set of time constants that can be assessed depending on the rate of specific processes governs the time-dependent behavior of systems. Additionally, comparing these time constants enables the assessment of which variable has a significant or minor impact on the system behavior. The reactor time constant ($\tau_p$), which represents the reactor response to a change in liquid flowrate, and the dead time ($t_d$), which measures the interval between a step change and the first response of the measured COD, were calculated using Equations (3) and (4), respectively [36].

$$t_d = t_1 - t_2 \qquad (3)$$

$$\tau_p = t_{63.2\%} - t_2 \qquad (4)$$

where $t_1$ is the time when the step change is made; $t_2$ represents the time when the measured COD first responds to the step change; $t_{63.2\%}$ is the time when the measured COD reaches 63.2% of its total final change. The results given in Table 5 and Figure 7 demonstrate that $t_d < \tau_p$ in the experiments, which indicates a tight and easy overall process control [36]. It was found that the dead time was connected to the step change direction in the liquid flow rate. For instance, when the liquid flow rate was reduced, the dead time decreased. These

findings are quite encouraging in terms of the adaptation and suitability of the reactor system for large-scale processes.

**Table 5.** Step-change in the liquid flow rate and HRT; dead time and time constant.

| Condition | Step Change | $t_1$ (h) | $t_2$ (h) | $t_{63.2\%}$ (h) | $t_d$ (h) | $\tau_p$ (h) |
|---|---|---|---|---|---|---|
| Liquid flow rate | 2.1–4.2 | 22 | 24 | 33.376 | 2 | 9.376 |
| Liquid flow rate | 4.2–2.1 | 48 | 48.5 | 56.848 | 0.5 | 6.848 |

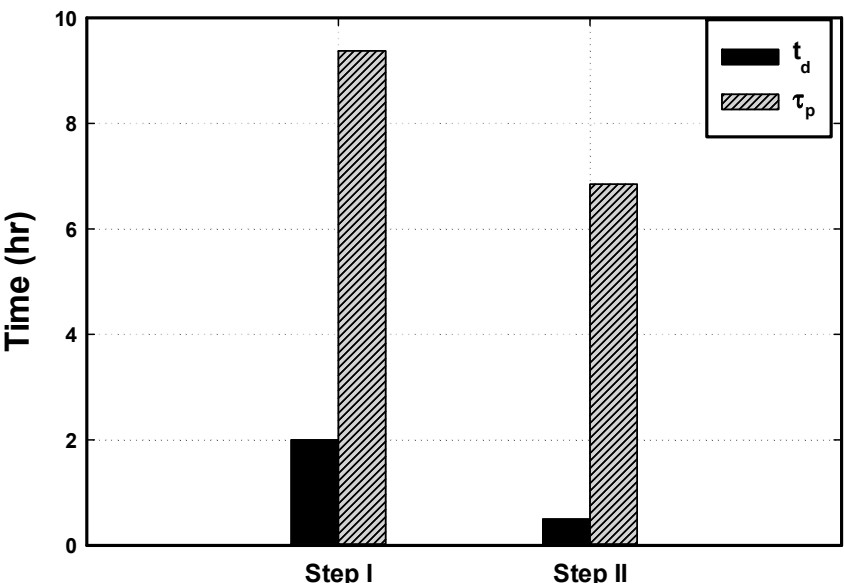

**Figure 7.** The dead time and reactor time constants for a step change experiment.

Although these continuous experiments were limited to studying only the effect of the liquid and air flow rates, other effects such as the initial COD and immobilized PVA particle size can be investigated in the future. A study of the continuous biotreatment of phenol in SBBR was performed by El-Naas et al. [21] who investigated the effect of the initial COD and PVA immobilized particle size on the biotreatment of phenol. It revealed that the rate of phenol reduction is stable for a longer period at high initial COD concentrations than at low initial COD concentrations, as it appears to decrease over time. Bacteria immobilized within the PVA particles may only have limited access to organic compounds at low COD concentrations, which can be explained by mass transfer limitations. In addition, the particle size of PVA has a significant effect on the continuous biotreatment of phenol, which is increased by particle size reduction, especially for high air flow rates.

The average COD for GTL process water treated in this study was found to be ~441 mg/L, which is considered to be purified water, as reported by Luis et al. [37]. Therefore, further pre/post-treatment is required. Table 6 shows GTL PW characteristics after treatment. Table 6 shows the treated GTL PW characteristics.

**Table 6.** The physical and chemical characteristics of GTL.

| Characteristic | GTL PW | Pretreated GTL PW | Treated GTL PW (Batch Experiments) |
|---|---|---|---|
| COD (mg/L) | 5000–7000 | 2000 to 4000 | ~441 |
| TOC (mg/L) | 1500–1700 | 700–1400 | ~154 |
| pH | 2.9 | 3.3 | 7.2 |

## 4. Conclusions

In a specially designed SBBR system, the biotreatment of organic contaminants in the GTL process was investigated. Using RSM, the second-order polynomial was found to be effective in forecasting COD reduction when three independent variables were used: initial COD concentration, pH, and PVA v%. All three factors had a considerable impact on the GTL PW biotreatment, and the maximum COD reduction for GTL PW was found to be 89% in batch experiments. Additionally, when applying the RSM optimizer for GTL PW, the optimal COD reduction was predicted to be 90% at the optimum condition of an initial COD of 2595 mg/L, PVA v% of 27%, and pH of 7.3. A continuous study was carried out to investigate the effect of the liquid flow rate and air flow rate on the organic removal from GTL PW. The results showed that when increasing the air flow rate from 2 to 3.3 $L_a/L_r$.min, the COD reduction and DO of GTL PW increased from 60.6% to 62%, 5.2 and 5.5 mg/L, respectively. This demonstrates that the difference between the two flow rates was insignificant. A step change experiment of the liquid flow rate from 2.1 to 4.2 mL/min and then back to 2.1 mL/min was performed. The system responded quickly to the change in liquid flow rate and returned to the initial COD level. This indicates that the system is highly stable and can easily recover. These findings are quite encouraging in terms of the adaptation and suitability of the reactor system for large-scale processes.

**Author Contributions:** Conceptualization, S.A.A. and M.H.E.-N.; Methodology S.A.A., R.S., M.M.B.-A. and M.H.E.-N.; Software, S.A.A.; Investigation, S.A.A., R.S., M.M.B.-A. and M.H.E.-N.; Data curation, S.A.A. and R.S.; Writing—original draft preparation, S.A.A.; Writing—review and editing, S.A.A., R.S., M.M.B.-A. and M.H.E.-N.; Supervision, M.M.B.-A. and M.H.E.-N.; Project administration, M.H.E.-N.; Funding acquisition, M.H.E.-N. All authors have read and agreed to the published version of the manuscript.

**Funding:** The authors would like to acknowledge the support of Qatar National Research Fund (a member of Qatar Foundation) through grant #NPRP100129170278.

**Data Availability Statement:** Not applicable.

**Acknowledgments:** The authors would like to acknowledge the help of Zulfa Al Disi from the biological science lab at Qatar University and Eng. Dan Cortes for his valuable technical assistance.

**Conflicts of Interest:** The authors declare no conflict of interest.

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
