# Peer review of "Optimization of the Biotreatment of GTL Process Water Using Pseudomonas aeruginosa Immobilized in PVA Hydrogel"

_processes, doi:10.3390/pr10122568_

Round 1

Reviewer 1 Report

The work is an average and routine type and cannot be accepted due to the following reasons. 1. This a routine type of work without any originality. 2. The novelty of the work is not established. 3. Quantitative information is not provided in the abstract. 4. The application and importance of work are not described. It should be published as University/Institute technical report.

Author Response

Please see attached responses to Reviewer 1.

Reviewer 2 Report

Please proof read before submission.  

1.      Line 69:  Rephrase, [The strains was effectively degraded the organic pollutants   …]

2.      Line 91-92, rephrase   … [the aim of this study aimed to investigate …   ]

3.      Line 100-101, Rewrite chemical formula giving particular care to subscripts. 

4.      Figures are not embedded in the sections where they are discussed in text.

5.      Figure 2 is labelled 2 times at page 9 and page 10

6.      Figure 4 on Page 11: . “The concentration of GTL PW in SBBR    …32 â—¦C; liquid flow rate = 2.1 ml/min.”

7.      How many times the experiment was conducted, any error bar?

8.      Figure 7: where is it.

9.      Figures are not placed in the text where they are discussed.

10.  Since the Figure labelling is wrong, therefore, the discussion of figures in the text is hard to understand.

11.  The conclusion part is not properly represented in abstract.

Author Response

Please see attached responses to Reviewer 2.

Reviewer 3 Report

The manuscript ‘Optimization of biotreatment of GTL process water using Pseudomonas aeruginosa immobilized in PVA hydrogel’ was carried out to optimize the COD reduction in Gas to liquid (GTL) process water by Response Surface Methodology (RSM), using the biodegradation process of Spouted Bed Bioreactor (SBBR) with Pseudomonas aeruginosa immobilized in polyvinyl alcohol (PVA) gel. The manuscript was well-written and meaningful, but there were some points need to be improved.

1.     Line 16 and 17, there lacked commas in numbers which was over 1,000. For example, ‘1,000’, ‘3,000’, and ‘2,595’.

2.     Line 60, I am confused about the word ‘treted’.

3.     It lacked error bars in some figures (e.g. Figure 4, and Figure 5).

Author Response

Please see attached responses to Reviewer 3.

Round 2

Reviewer 1 Report

The manuscript's quality has been improved. I recommend its acceptance for publication in its present form.